# Effects of an Explosion-Proof Wall on Shock Wave Parameters and Safe Area Prediction

Dingjun Xiao [1,2,*], Wentao Yang [2,3], Moujin Lin [2], Xiaoming Lü [4], Kaide Liu [1], Jin Zhang [2], Xiaoshuang Li [2] and Yu Long [2]

1　Shaanxi Key Laboratory of Safety and Durability of Concrete Structures, Xijing University, Xi'an 710123, China; liukaide2006@163.com
2　School of Environment and Resource, Southwest University of Science and Technology, Mianyang 621010, China; 15775965983@163.com (W.Y.); lmj2012pt@163.com (M.L.); zhangjinswust@163.com (J.Z.); 13198097226@163.com (X.L.); ly17743299952@163.com (Y.L.)
3　Chengdu Institute of Urban Safety and Emergency Management, Chengdu 610011, China
4　Mechanical Engineering Research Institute, Xi'an 710123, China; lvxiaoming2006@163.com
*　Correspondence: dingjun@swust.edu.cn; Tel.: +86-13990176379

**Abstract:** To study the influences of an explosion-proof wall on shock wave parameters, an air explosion protection experiment was performed, the time history of shock wave pressure at different positions before and after the explosion-proof wall was established, and the characteristics of shock wave impulse and dynamic pressure were analyzed. The explosion-proof working conditions of five different diffraction angles were simulated and analyzed using Autodyn software(2019R3). Results indicated the following findings. The explosion-proof wall exerted an evident attenuation effect on the explosion shock wave, but considerable pressure still existed at the top of the explosion-proof wall. Overpressure behind the wall initially increased and then decreased. The larger the diffraction angle, the faster the attenuation speed of the diffraction overpressure of the shock wave in the air behind the wall. The history curve of shock wave pressure exhibited an evident bimodal structure. The shock wave diffraction of the wall made the shock wave bimodal structure behind the wall more prominent. The characteristics of the bimodal structure behind the wall (the interval time of overpressure peak $\Delta t$ was less than the normal phase time of the diffracted shock wave $T^+$) caused the shock wave impulse to stack rapidly, significantly improving its damage capability. The peak value of dynamic pressure on the oncoming surface was approximately two times the peak value of overpressure, and the inertia of air molecules resulted in a longer positive duration of dynamic pressure than overpressure. The maximum overpressure on the ground behind the explosion-proof wall appeared at approximately two times the height of the explosion-proof wall, while the maximum overpressure in the air behind the explosion-proof wall appeared at approximately one times the height of the explosion-proof wall. The relatively safe areas on the ground and in the air behind the wall were approximately 4–4.5 times and 3.5–4 times the height of the explosion-proof wall, respectively.

**Keywords:** air explosion; diffraction angle; overpressure; impulse; dynamic pressure; overpressure prediction; safe area

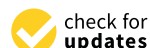



## 1. Introduction

As a modern field protection facility, an explosion-proof wall can effectively isolate the explosion source and protect personnel, considerably reducing the damage and destructive effects of explosive shock waves and shrapnel on important military targets and personnel [1–3], thus improving the survivability of military personnel. As a major wartime damage element, the blast shock wave will be reflected near the ground, superimposed with the incident shock wave, and act on the blast-facing surface of the blast wall, causing the blast wall to be severely damaged or even overturned [4]. Simultaneously, the explosion shock wave diffracts through the upper and both sides of the wall after coming in contact

with the explosion-proof wall, changing the shock wave pressure flow field behind the wall and even forming a higher overpressure in a specific area behind the wall [5,6], causing damage to the evading personnel. Therefore, studying the influences of an explosion-proof wall on the explosion shock wave flow field and the law of shock wave parameters is highly significant to evaluate the protective performance of an explosion-proof wall [7–10].

The anti-overturning and stability of explosion-proof walls have achieved certain progress in domestic and foreign research. Zhang [11] and Scherbatiuk K [3,4,6] established a finite element model of the explosion load and the response characteristics of an explosion-proof wall, proposed an analytical model of such a wall based on the rotation of a rigid body, and analyzed the response characteristics of an explosion-proof wall on the basis of the displacement time curve measured in the experiment. Explosion-proof walls of different materials exert varying blocking effects on the explosion shock wave. When the explosion shock wave passes through a concrete wall [12], it will produce an evident circulation phenomenon, which exerts a significant effect on the flow field behind the wall; the buffer effect of the soft medium explosion-proof wall of the water pack is evident [13,14]. The environment of an explosion-proof wall also affects shock wave overpressure: when explosives detonate in different environments, such as air, rigid ground, and sandy ground, they affect the distribution of overpressure in different directions [15]. The different geometric dimensions of an explosion-proof wall affect the diffraction angle and also exert a considerable influence on the propagation of shock waves [16–18]. Zhu and Mu [19,20] determined that the maximum value of the diffraction overpressure behind the wall is related to the value of the diffraction angle. However, the relationship between the diffraction angle and the shock wave diffracted flow field has not yet been studied in detail. In summary, the environmental and geometric parameters of an explosion-proof wall exhibit an effect on its anti-explosive performance, and thus, the shock wave overpressure [21,22], specific shock volume, and dynamic pressure of such a wall [23,24] should be comprehensively analyzed when evaluating its anti-explosive performance. Most studies have focused only on one of the three parameters and cannot completely investigate the impact protection performance of an explosion-proof wall. The choice of an appropriate material dynamic model has a significant impact on the results of numerical calculations. Ma [25] proposed an improved Johnson–Holmquist–Beissel model that can effectively predict the projectile penetration behavior of brittle materials. A new modeling strategy with inclusions and an FDEM method is also proposed to study the dynamic response and fracture behavior of geomaterials [26]. In this paper, the compaction nonlinear state equation is used to describe the effect of blast walls in blast loading.

In the current study, two types of sensors were arranged on the ground and in the air behind an explosion-proof wall, considerably covering the hazard area of the shock wave behind the explosion-proof wall for personnel and equipment. The calculation of shock wave overpressure, impulse, and dynamic pressure at each measuring point was analyzed in detail. The diffraction angle range was set to 22°–44°. The corresponding explosion distance can be determined in accordance with its geometric relationship. The pressure variation law of diffracted shock waves from varying angles at different positions of the explosion-proof wall was studied through an experiment and numerical simulation to identify the parameters related to explosion safety protection and to provide a reference for designing the size and structure of an explosion-proof wall, safely avoiding the damage caused by shock waves.

## 2. Experiment

### 2.1. Shock Wave Testing System

The shock wave pressure sensors used in this work were 10 wall-type PCB shock wave pressure sensors with a minimum resolution of 0.10 kPa and 10 free-field PCB shock wave pressure sensors with a minimum resolution of 0.07 kPa. The shock wave data acquisition instrument adopted a Chengdu Tytest Blast-PRO shock wave tester with a sampling rate of 4 MHz and a Jiangsu Donghua DH5960 data acquisition instrument with a sampling

rate of 20 MHz. The analysis system was primarily composed of the PC terminal and its supporting software. The shock wave test system is shown in Figure 1.

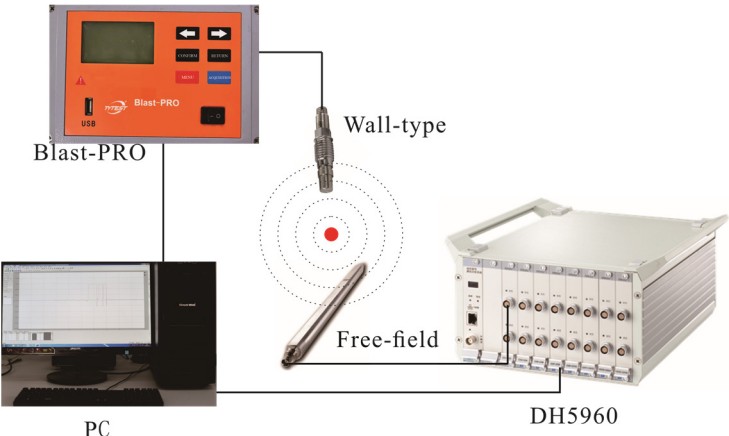

**Figure 1.** Shock wave testing system.

### 2.2. Experimental Design

Two sets of explosion-proof wall anti-explosion experiments were established. The trinitrotoluene (TNT) equivalent was 1.5 kg, and the explosion height was 1.2 m. The proportional distance between the explosion-proof wall and the explosion source, the distance, and the corresponding relationship of the diffraction angle $\alpha$ are provided in Table 1 and Figure 2. The experimental explosion-proof unit consisted of five individual sand-filled blast walls with a single blast wall geometry of 1 m × 1 m × 2.1 m.

**Table 1.** Relationship between proportional distance and explosion distance.

| Test Group | Proportional Distance/m·kg$^{-1/3}$ | Burst Distance/m | Diffraction Angle $\alpha$/° |
|:---:|:---:|:---:|:---:|
| 1-1 | 1.26 | 1.44 | 32 |
| 1-2 | 1.48 | 1.69 | 28 |

1~10–The gauge of overpressure on ground

11~20–The gauge of overpressure on air

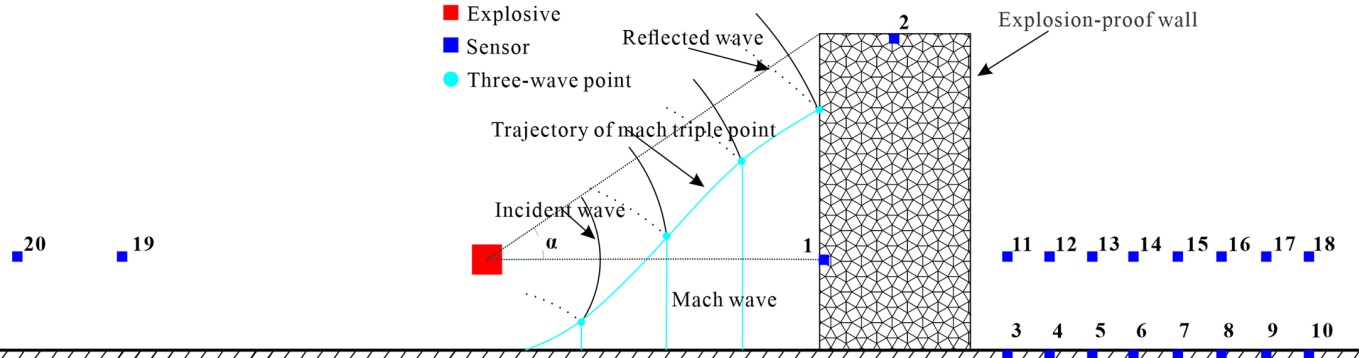

**Figure 2.** Diagram showing the layout of shock wave measuring points.

The types of deployed sensors were mostly wall-type shock wave sensors (measurement points 1–10) and free-field shock wave sensors (measurement points 11–20). The height of the free-field pressure sensors from the ground was 1.2 m. The shock wave sensors were arranged at 1, 2, 3, 4, 5, 6, 7, and 8 m from the back of the explosion-proof wall. The literature [19] pointed out that the maximum overpressure behind the wall occurs within

1.5–2.5 times the height of the explosion-proof wall behind the wall. The distance between measuring points 19 and 20 and the charge is consistent with the horizontal distance from measuring points 5 (13) and 9 (17) to the explosion source, respectively. The outer layer of the explosion-proof wall used in the experiment was a reinforced mesh frame, an inner layer lined with a geotextile pocket protective sheet, and an inner layer filled with an energy-absorbing material (i.e., sand). The height of the wall was 2.1 m, the width was 5 m, and three soil samples were collected. The measured physical parameters are listed in Table 2. The experimental explosion-proof unit consisted of five individual sand-filled blast walls with a single blast wall geometry of 1 m × 1 m × 2.1 m.

**Table 2.** Density and moisture content of sand.

| Sand | Density/g·cm$^{-1/3}$ | Water Content/% |
|---|---|---|
| 1 | 2.017 | 35.16 |
| 2 | 2.033 | 31.06 |
| 3 | 2.033 | 33.10 |

### 2.3. Layout of Measuring Points

This explosion experiment was performed in an open field. The sensor should be level when fixing the air free-field sensor. In accordance with the actual situation, the shock wave will be diffracted on the top, left, and right sides of the explosion-proof wall. To ensure the integrity of the shock wave signal test, the plane part of the sensor was directed upward, and the sensor was pointed toward the grain tested and on the same horizontal plane as the center of the explosion source. Meanwhile, to ensure the stability of the free-field sensor in the air, a fixed sandbag was placed at the bottom of the fixed support and secured with a nylon rope. Finally, the on-site layout is depicted in Figure 3.

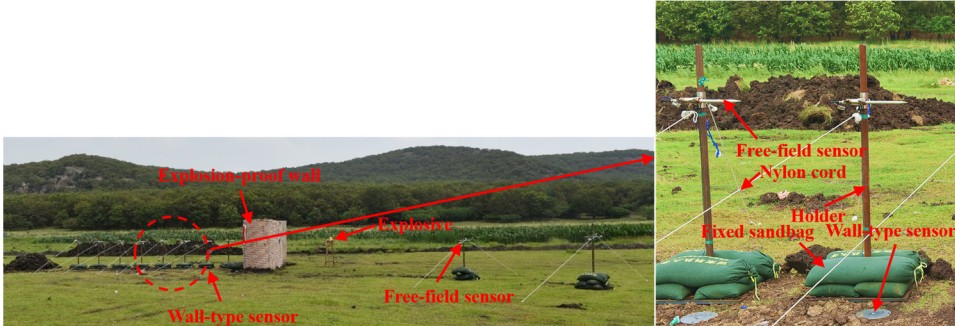

**Figure 3.** Physical layout of shock wave measure points.

## 3. Results and Analysis

### 3.1. Shock Wave Overpressure

Considering the large number of measuring points in this experiment, the pressure time history curves of some typical measuring points in Experiment 1-1 were selected for explanation below in order to avoid this paper being too lengthy, as shown in Figure 4.

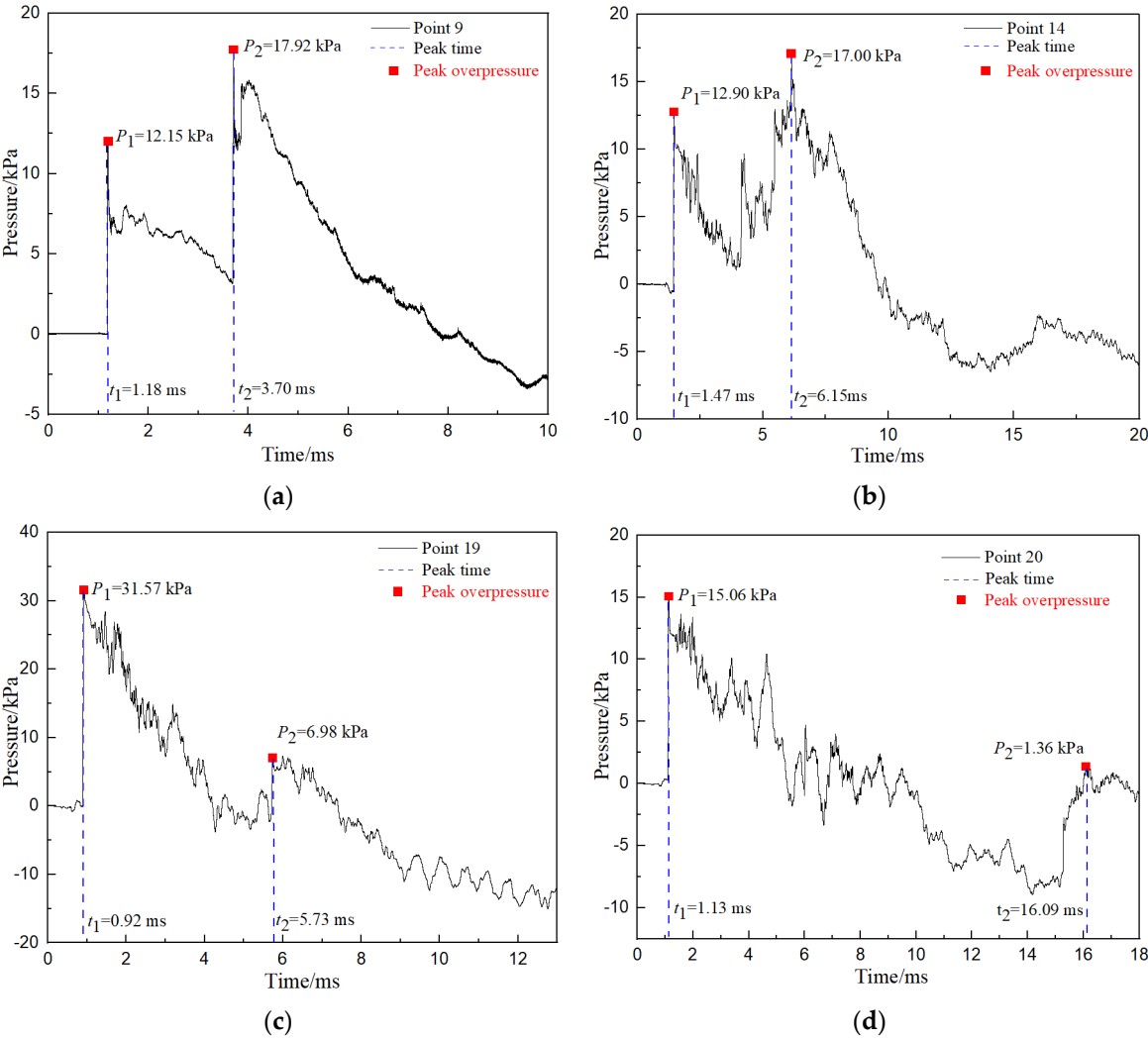

**Figure 4.** Diffraction angle 32°: time history curve of typical shock wave pressure: (**a**) Point 9; (**b**) Point 14; (**c**) Point 19; (**d**) Point 20.

Figure 4 shows the shock wave pressure time history curves of several measuring points in Experiment 1-1, where $P_1$ is the peak pressure of the diffraction wave; $P_2$ is the peak pressure of the reflected wave; and $t_1$ and $t_2$ are the arrival time of the diffracted and reflected waves, respectively. The aforementioned measuring points were located on the ground (9) and in the air (14) of the back explosion surface, and in the air of the front explosion surface (19,20). The diffraction overpressure peak, reflection overpressure peak, and the corresponding time of the two overpressure peaks at each measuring point were provided. The shock wave pressure time history curve of the two measuring points in the air of the blast-facing surface under the action of a 1.5 kg TNT air explosion is similar to the ideal shock wave pressure time history curve. The peak incident overpressure of the shock wave at measuring points 19 and 20 was 31.57 kPa and 15.06 kPa, respectively. In accordance with the empirical formula of Hengrych J [27] (Formula (1)), the shock wave overpressure at measuring point 19 was 34.91 kPa and that at measuring point 20 was 14.56 kPa, which are basically the same as the measured shock wave overpressure, demonstrating the reliability of this test. Evident "double peak" waveforms can be observed in the shock wave pressure time history curve of each measuring point behind the wall, and this time history curve differs from the typical air free-field shock wave pressure time history curve. The reflected wave overpressure appeared when the diffracted wave attenuation process did not enter

the negative phase. The shock wave parameters of the second experiment are provided in Tables 3 and 4.

$$\Delta P_m = \begin{cases} 1407.17/\overline{R} + 553.97/\overline{R}^2 - 35.72/\overline{R}^3 + 0.625/\overline{R}^4, 0.05 \le \overline{R} \le 0.3 \\ 619.38/\overline{R} - 32.62/\overline{R}^2 - 213.24/\overline{R}^3, 0.3 \le \overline{R} \le 1 \\ 66.2/\overline{R} + 405/\overline{R}^2 + 328.8/\overline{R}^3, 1 \le \overline{R} \le 10 \end{cases} \tag{1}$$

where $\Delta P_m$ is the peak overpressure, kPa; $\overline{R} = R/\sqrt[3]{W}$; $\overline{R}$ is the proportional distance, m·kg$^{-1/3}$; $R$ is the blast distance, m; and $W$ is the TNT equivalent, kg.

**Table 3.** Diffraction angle 32°: shock wave pressure parameters.

| Measuring Points | Peak Value of Diffraction Wave/kPa | Peak Value of Reflected Wave/kPa | $\Delta t$/ms |
|---|---|---|---|
| 1 | 3845.50 | 2628.31 | 0.18 |
| 2 | 254.38 | 28.73 | 1.23 |
| 3 | 16.47 | 1.51 | 6.05 |
| 4 | 11.23 | 13.30 | 3.92 |
| 5 | 11.25 | 13.20 | 2.91 |
| 6 | 10.46 | 14.30 | 3.28 |
| 7 | 18.51 | 19.69 | 3.20 |
| 8 | 13.89 | 19.00 | 2.92 |
| 9 | 12.15 | 17.92 | 2.52 |
| 10 | 8.49 | 19.53 | 2.89 |
| 11 | 22.49 | 10.65 | 6.46 |
| 12 | 20.62 | 10.51 | 6.18 |
| 13 | 20.67 | 19.39 | 4.84 |
| 14 | 12.90 | 17.00 | 4.68 |
| 15 | 12.87 | 22.75 | 4.39 |
| 16 | 10.03 | 13.23 | 3.60 |
| 17 | 8.81 | 13.34 | 3.36 |
| 18 | 6.87 | 13.18 | 3.13 |
| 19 | 31.57 | 6.98 | 4.81 |
| 20 | 15.06 | 1.36 | 14.96 |

Note: Interval between the reflected and diffracted waves, $\Delta t = t_2 - t_1$.

**Table 4.** Diffraction angle 28°: shock wave pressure parameters.

| Measuring Points | Peak Value of Diffraction Wave/kPa | Peak Value of Reflected Wave/kPa | $\Delta t$/ms |
|---|---|---|---|
| 1 | 3754.10 | 2421.07 | 0.24 |
| 2 | 271.20 | 169.21 | 2.97 |
| 3 | - | - | - |
| 4 | 9.43 | - | - |
| 5 | 8.50 | 11.32 | 4.46 |
| 6 | 10.66 | 10.45 | 3.19 |
| 7 | 13.86 | 14.29 | 3.62 |
| 8 | 13.97 | 14.89 | 3.43 |
| 9 | 10.67 | 15.60 | 2.93 |
| 10 | 9.27 | 14.12 | 2.85 |
| 11 | 20.92 | 10.09 | 6.68 |
| 12 | 18.53 | 11.05 | 5.76 |
| 13 | 19.79 | 20.14 | 4.86 |
| 14 | 14.37 | 15.35 | 4.74 |
| 15 | 13.06 | 18.99 | 4.79 |
| 16 | 8.52 | 12.64 | 3.98 |
| 17 | 8.31 | 11.37 | 3.72 |
| 18 | 7.84 | 10.26 | 3.53 |
| 19 | 25.97 | 10.46 | 5.67 |
| 20 | 15.83 | 6.07 | 7.67 |

Tables 3 and 4 show that the explosion-proof wall exerted a significant effect on reducing the shock wave of the explosion caused by 1.5 kg of TNT. The overpressure of the front explosion surface of the two actual explosion experiments was 3845.50 kPa and 3754.10 kPa. A large overpressure remained when the top of the explosion-proof wall was not protected, and the shock wave overpressure was lower on the back side of the explosion-proof wall. The overpressure distribution behind the explosion-proof wall is shown in Figures 5 and 6. The time relationship between the reflected and diffracted waves is illustrated in Figure 7.

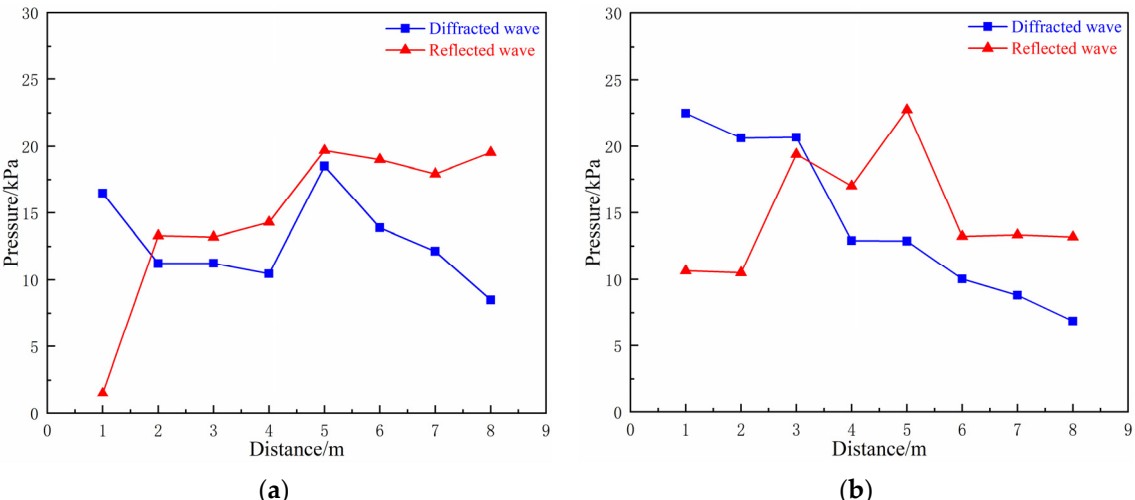

**Figure 5.** Diffraction angle 32°: overpressure distribution behind the wall: (**a**) ground; (**b**) air.

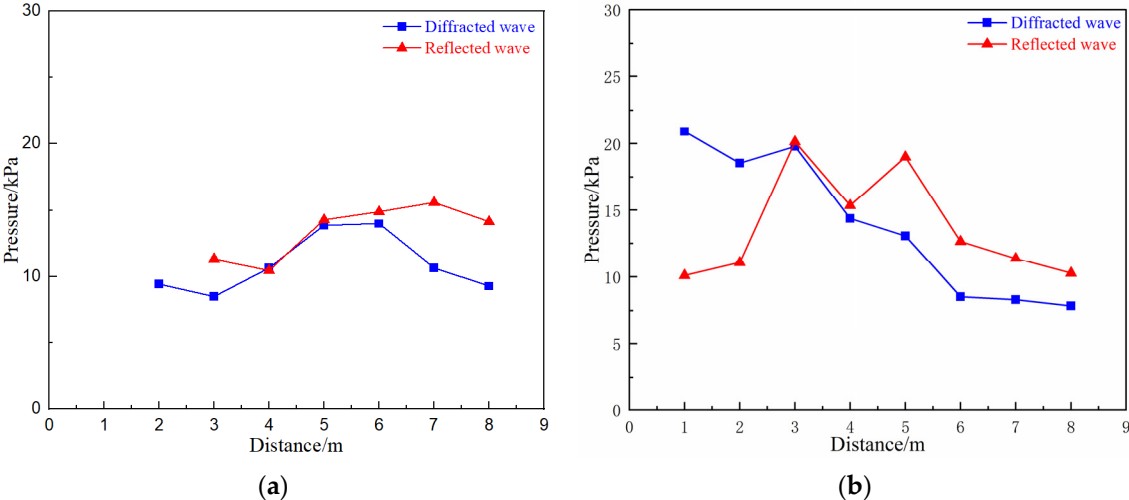

**Figure 6.** Diffraction angle 28°: overpressure distribution behind the wall: (**a**) ground; (**b**) air.

Figure 5 suggests that the diffracted and reflected waves can hardly reach the bottom behind the explosion-proof wall when the diffraction angle was 32°. Moreover, the diffracted overpressure was 16.47 kPa and the reflected overpressure was 1.51 kPa at measuring point 3. In accordance with the TNT air explosion conditions and the propagation path of the shock wave bypassing the explosion-proof wall, the diffraction overpressure was primarily caused by the explosion shock wave diffracting from the top of the explosion-proof wall, and the overpressure reflected was mostly formed by the shock wave being diffracted and reflected along the two sides of the explosion-proof wall. The overpressure reflected by the ground measure point behind the wall would be greater than the diffraction overpressure after approximately one times the height of the explosion-proof wall. In the subsequent measurement points, the reflected overpressure would be greater than the

diffraction overpressure, and then the maximum value of 19.69 kPa would be reached at a distance of 5 m (approximately 2.4 times the height of the explosion-proof wall). Here, the diffraction overpressure also reached the maximum value of 18.51 kPa, exhibiting a trend of initially increasing and then decreasing. In the air of the back explosion surface, diffraction overpressure gradually decreased with an increase in the distance from the explosion-proof wall. The maximum diffraction overpressure was 22.49 kPa at 1 m behind the wall, and reflection overpressure initially increased and then decreased. The maximum reflection overpressure was 22.75 kPa at 5 m away from the explosion-proof wall. At 3 m behind the wall (approximately 1.5 times the height of the explosion-proof wall), air reflection overpressure began to become greater than diffraction overpressure. As shown in Figure 6, the shock wave overpressure on the ground behind the wall initially increased and then decreased when the diffraction angle was small, with a slight difference between diffraction overpressure and reflection overpressure. In the measuring points after approximately two times the height of the explosion-proof wall, reflection overpressure was always greater than diffraction overpressure, and the maximum diffraction overpressure was 13.97 kPa at 5–6 m away from the explosion-proof wall (approximately 2.4–2.6 times the height of the explosion-proof wall). In the air above the back explosion surface, diffraction overpressure also decreased as the distance from the explosion-proof wall increased. The maximum diffraction overpressure was 20.92 kPa at 1 m behind the wall. Figures 5b and 6b show that as the diffraction angle decreased, the attenuation rate of diffraction overpressure ($n = (P_{11} - P_{18})/P_{11}$) in the air behind the wall decreased by 6.9%, indicating that the greater the diffraction angle, the faster the attenuation rate of diffraction overpressure in the air behind the wall increased with the distance from the explosion-proof wall. Reflection overpressure initially increased and then decreased. The maximum reflection overpressure was 20.14 kPa at 3 m from the explosion-proof wall (approximately 1.5 times the height of the explosion-proof wall). Simultaneously, reflection overpressure in the air began to become greater than diffraction overpressure, and reflection overpressure was always greater than diffraction overpressure.

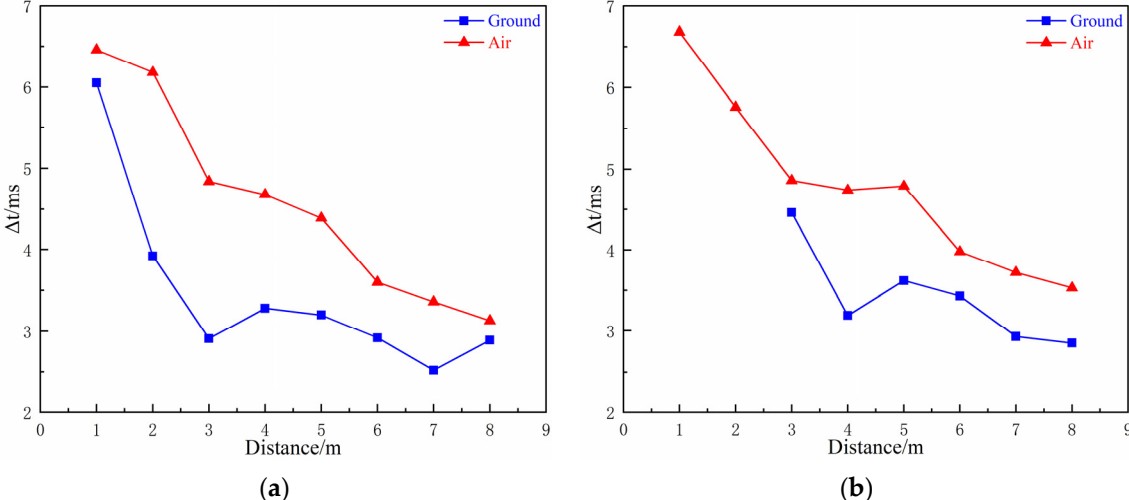

(**a**)　　　　　　　　　　　　　　　　　　　　　(**b**)

**Figure 7.** Interval time between the incident and reflected waves behind the wall: (**a**) diffraction angle 32°; (**b**) diffraction angle 28°.

　　Figures 4–6 suggest that the TNT explosion shock wave will exhibit two more evident overpressure peaks in the pressure time history curve. Figure 7 shows that the interval time between diffraction overpressure and reflection overpressure in the pressure time history curve gradually decreased with an increase in the distance from the explosion-proof wall. This interval time is also a key factor that affects the destructive effect of the explosion shock wave. If the interval time is less than the positive phase time of the diffraction shock wave,

then the two overpressure peaks are in the stage of rapid impulse rise, and the impulse is superimposed rapidly, exerting a strong destructive effect.

### 3.2. Overpressure Impulse

(1)    Impulse

In addition to the shock wave overpressure describing the power and destructive capability of the ammunition when it explodes, the impulse is also one of the important indicators. Its positive impulse is expressed as

$$I^+ = \int_{t_1}^{T^+} [P(t) - P_0]dt \tag{2}$$

where $I^+$ is the impulse of the positive shock wave, Pa·s; $t_1$ is the arrival time of the diffraction wave, ms; $T^+$ is the normal phase duration of the diffracted wave, ms; $P(t)$ is the pressure at this location as a function of time, kPa; and $P_0$ is the ambient pressure, kPa. The pressure time history curve obtained in the experiment was based on atmospheric pressure.

The impact of ground reflected waves on the explosion impulse cannot be ignored. If they are located below the three-wave point traces, then the ground reflected waves will obviously affect the pressure peak, the explosion directly generated by the incident wave, and will be superimposed and enhanced. If located above the three-wave point traces, the incident wave and the ground reflected waves will be significantly separated, weakening the pressure peak. The explosion-proof wall was installed in a field explosion; thus, the shock wave reflection situation was complicated and evidently different from the attenuation history in the free-field pressure time history curve of an unobstructed air explosion. The overpressure of the reflected wave at some positions was greater than that of the diffraction. Therefore, the explosive impulse studied in this work was not only the positive impulse phase a of the diffracted wave attenuation history but also included the negative impulse phase b between the diffracted and reflected waves and impulse phase c of the reflected wave attenuation history. If the reflected wave arrives before the pressure of the diffracted wave enters the negative phase, then phase b will not occur. If the overpressure of the reflected wave has a negative value, then only the positive impulse a of the diffracted wave is calculated, as shown in Figure 8.

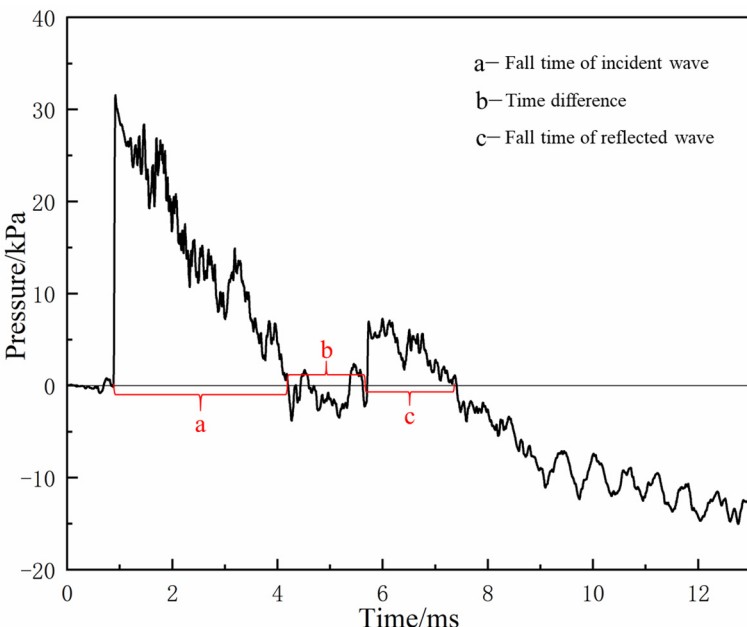

**Figure 8.** Impulse calculation rules.

(2)    Characteristics of the impulse curve at different positions

Figure 9 shows the impulse time history curve at several positions, illustrating the evident difference between the arrival of the reflected wave before and after the pressure of the diffracted wave decays into the negative phase. In Figure 9, a and b illustrates the impulse curves of the two measuring points behind the wall. The reflected wave reached the measuring point before entering the negative phase in the attenuation process of the diffracted wave. The impulse increased rapidly after the arrival of the reflected wave. For example, the impulse of the diffracted wave at measuring point 9 was 15.17 Pa·s. The impulse continued to increase with the arrival of the reflected wave, and the maximum impulse was 43.19 Pa·s. At measuring point 14, the impulse of the diffracted wave was 30.08 Pa·s and the maximum impulse was 57.13 Pa·s. By contrast, for c and d in Figure 9, the reflected wave arrived after the diffracted wave entered the negative phase, indicating that its impulse decreased. In this case, the impulse of the diffracted wave was generally considerably greater than that of the reflected wave. For example, the impulse of the diffracted wave at measuring point 19 was 49.41 Pa·s, and the maximum impulse was 54.65 Pa·s. At measuring point 20, the impulse of the diffracted wave was 35.66 Pa·s, which was also the maximum impulse. The peak overpressure of the reflected wave was small because the reflected wave arrived late, and a continuous impulse rising phase did not occur. Thus, the impulse at the measuring point dropped rapidly.

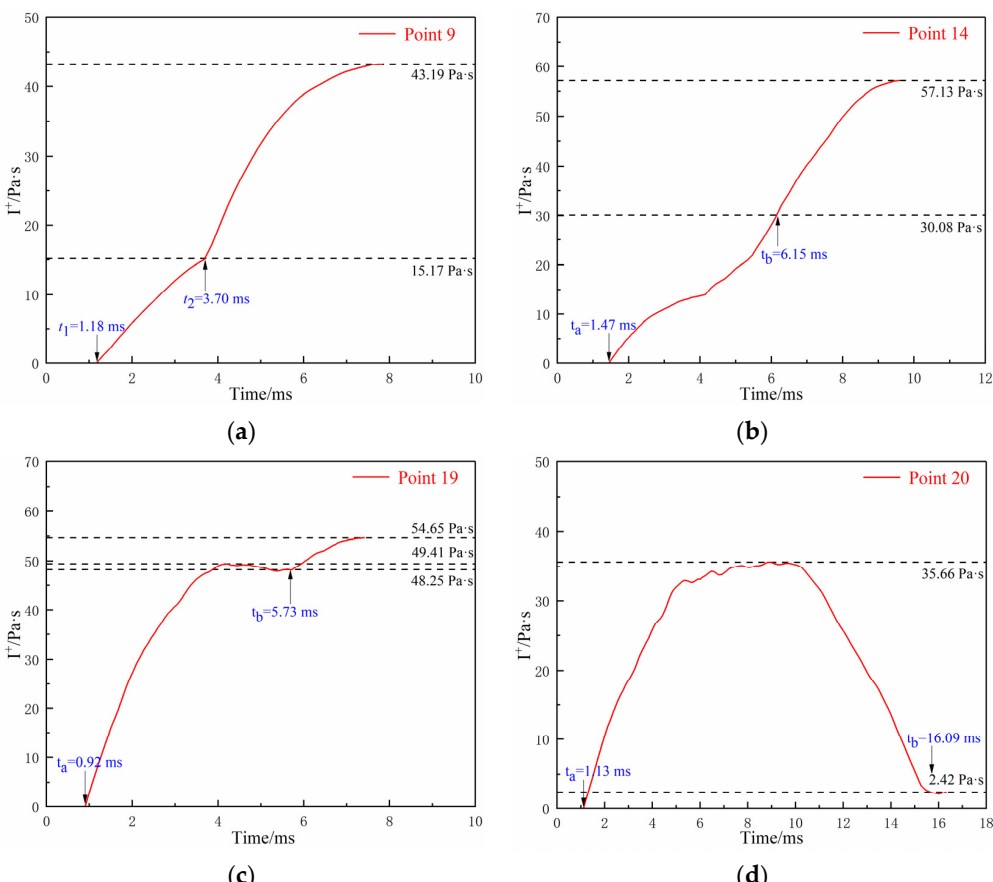

**Figure 9.** Diffraction angle 32°: typical impulse time history curve: (**a**) Point 9; (**b**) Point 14; (**c**) Point 19; (**d**) Point 20.

Tables 5 and 6 provide the positive phase impulse of the diffracted wave, the cumulative impulse of the reflected wave before reaching the negative phase, and the maximum impulse of the reflected wave before entering the negative phase after the arrival of the reflected wave in Experiments 1-1 and 1-2. In the two experiments, the maximum impulse of most measuring points was equal to the accumulated impulse before the reflected wave reached the negative phase. A considerable difference in value was noted compared with

the positive phase impulse of the diffracted wave because the reflected wave of the explosion shock wave arrived before the diffracted wave entered the negative phase. Thus, the impulse history curve continued to rise. At measuring points 2, 18, and 19, the maximum impulse of the reflected wave before entering the negative phase was close to the positive impulse of the diffracted wave, or even less than the diffracted impulse. This condition was attributed to the reflected wave reaching the measuring point after the diffracted wave entered the negative phase. The negative phase impulse offset a large amount of reflected impulse.

**Table 5.** Diffraction angle 32°: impulse.

| Measuring Points | Positive Impulse of Diffraction Wave/Pa·s | Cumulative Impulse of Negative Reflection Wave Front/Pa·s | Maximum Impulse/Pa·s |
|---|---|---|---|
| 1 | 97.21 | 217.52 | 217.52 |
| 2 | 44.11 | 5.86 | 47.16 |
| 3 | 16.83 | 57.79 | 57.79 |
| 4 | 15.27 | 49.74 | 49.74 |
| 5 | 15.08 | 49.21 | 49.21 |
| 6 | 15.17 | 45.87 | 45.87 |
| 7 | 18.86 | 54.19 | 54.19 |
| 8 | 13.94 | 46.39 | 46.39 |
| 9 | 15.17 | 43.19 | 43.19 |
| 10 | 19.67 | 60.69 | 60.69 |
| 11 | 22.97 | 47.23 | 47.23 |
| 12 | 20.71 | 49.12 | 49.12 |
| 13 | 23.72 | 64.77 | 64.77 |
| 14 | 30.08 | 57.13 | 57.13 |
| 15 | 12.21 | 62.71 | 62.71 |
| 16 | 11.31 | 47.99 | 47.99 |
| 17 | 10.81 | 44.98 | 44.98 |
| 18 | 9.49 | 39.84 | 39.84 |
| 19 | 49.41 | 54.65 | 54.65 |
| 20 | 35.66 | 2.42 | 35.66 |

**Table 6.** Diffraction angle 28°: impulse.

| Measuring Points | Positive Impulse of Diffraction Wave/Pa·s | Cumulative Impulse of Negative Reflection Wave Front/Pa·s | Maximum Impulse/Pa·s |
|---|---|---|---|
| 1 | 189.27 | 338.91 | 338.91 |
| 2 | 35.14 | −39.16 | 35.14 |
| 3 | - | - | - |
| 4 | 11.57 | - | - |
| 5 | 14.34 | 41.47 | 41.47 |
| 6 | 12.93 | 33.03 | 33.03 |
| 7 | 20.23 | 50.97 | 50.97 |
| 8 | 17.69 | 45.94 | 45.94 |
| 9 | 16.83 | 42.15 | 42.15 |
| 10 | 13.62 | 34.43 | 34.43 |
| 11 | 23.32 | 50.35 | 50.35 |
| 12 | 14.24 | 49.90 | 49.90 |
| 13 | 16.55 | 68.22 | 68.22 |
| 14 | 16.44 | 60.28 | 60.28 |
| 15 | 13.00 | 55.78 | 55.78 |
| 16 | 10.16 | 39.16 | 39.16 |
| 17 | 9.03 | 37.45 | 37.45 |
| 18 | 7.76 | 34.79 | 34.79 |
| 19 | 46.00 | 47.86 | 47.86 |
| 20 | 30.59 | 33.62 | 33.62 |

Figure 10 shows the impulse comparison chart of the air and ground measurement points in the two experiments at different distances behind the explosion-proof wall and the average value of the impulse of the air and ground measurement points. When the diffraction angle was 32°, the mean impulse values of the air and ground measuring points were basically equal, and the impulse values at the two measuring points at different heights were close at the same horizontal distance from the explosion source. When the diffraction angle was 28°, a certain difference was noted in the mean impulse value between the air and ground measuring points. The recorded data show that the impulse values recorded by the air and ground measuring points gradually approached each other with an increase in distance. The maximum impulse in the air behind the wall of the two experiments was 3–5 m behind the wall (approximately 1.5–2.4 times the height of the wall). These values are consistent with the diffraction and reflection overpressure in b of Figures 5 and 6. At the two diffraction angles, the average air impulse was greater than the ground average impulse, and the change in diffraction angle exerted a minimal effect on the air diffraction impulse.

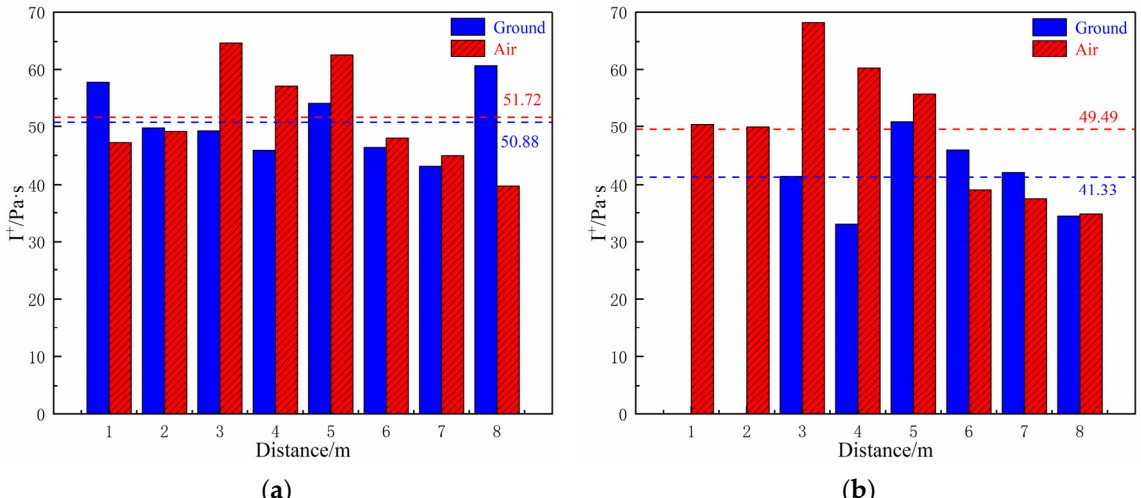

**Figure 10.** Comparison chart of air and ground impulse; (**a**) diffraction angle 32°; (**b**) diffraction angle 28°.

### 3.3. Shock Wave Dynamic Pressure

With the rapid development of new high-energy weapons and their battlefield applications, in contrast with shock wave overpressure, diffraction pressure and reflected pressure last only for a short time (typically much less than 1 s). However, dynamic pressure may last longer (up to 2–3 s), and the damaging effect of dynamic pressure becomes more prominent. At present, dynamic pressure measurement experiences many technical difficulties. The literature [23] asserted that air can be regarded as a hot complete gas when shock wave overpressure was less than 4 MPa and wave front temperature was less than 2000 K. The current study used the literature [24] as a reference to present a method for calculating typical air impact fluctuating pressure under an ideal explosion condition (i.e., the air is a complete hot gas, the ground has no thermal layer, and a large amount of dust is absent from the air).

$$q_s = 2.45 \times \frac{\Delta p_m^2}{7.2 + \Delta P_m} \tag{3}$$

where $q_s$ is the peak value of dynamic pressure, kPa.

Through calculation, Table 7 shows the dynamic pressure peaks at several measuring points on the blasting face of the two experiments.

**Table 7.** Dynamic pressure peak.

| Measuring Points | Dynamic Pressure Peak/kPa | |
|---|---|---|
| | **32°** | **28°** |
| 1 | 9403.87 | 9179.94 |
| 19 | 62.98 | 49.81 |
| 20 | 24.96 | 26.67 |

The closer the distance between the dynamic pressure peak on the blasting face of the explosion-proof wall and the explosive, the greater the dynamic pressure peak. In accordance with the overpressure time history curve, each time corresponds to the overpressure of a wave front, and the dynamic pressure time history curve can be drawn as shown in Figures 11 and 12.

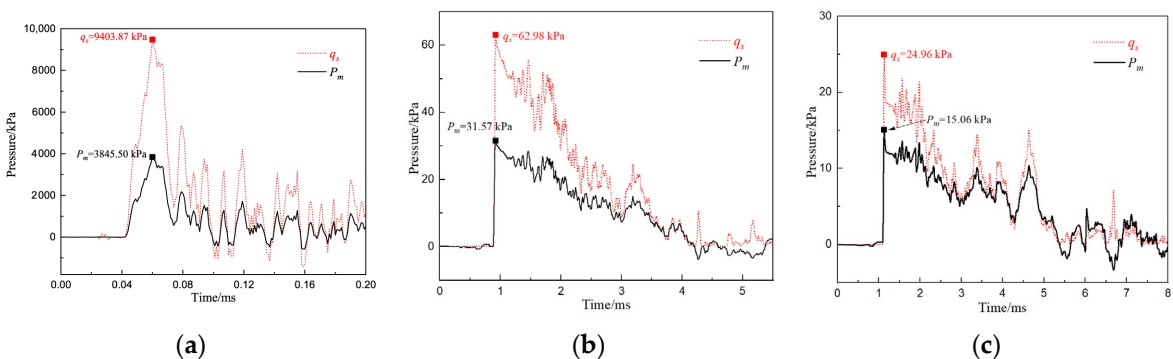

**Figure 11.** Diffraction angle 32°: overpressure and dynamic pressure: (**a**) Point 1; (**b**) Point 19; (**c**) Point 20.

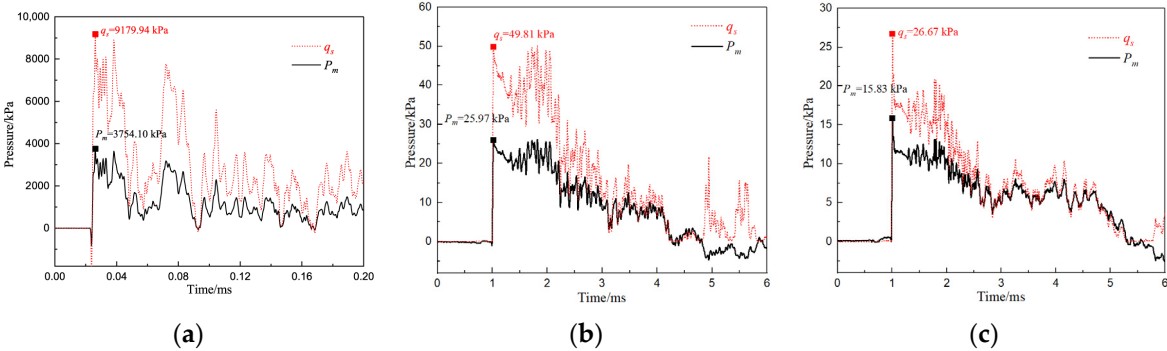

**Figure 12.** Diffraction angle 28°: overpressure and dynamic pressure: (**a**) Point 1; (**b**) Point 19; (**c**) Point 20.

Table 7 and Figures 11 and 12 show that the peak value of dynamic pressure at measuring points 1, 19, and 20 on the explosion-facing surface was approximately two times the peak overpressure. From the overpressure and dynamic pressure time history curves, the dynamic pressure peak decreased with a decrease in the overpressure peak in the positive phase of overpressure. In the two experiments, when shock wave overpressure at measuring points 19 and 20 entered the negative phase, the air molecules behind the shock wave front possessed inertia and would not immediately drop into the negative phase. Measuring point 1 was located on the explosion-facing surface of the explosion-proof wall. It would not enter the negative phase immediately with large shock wave reflection overpressure, and impact fluctuating pressure would also have a long positive phase duration.

### 4. Numerical Simulation

#### 4.1. Simulation Design

The diffraction angle range of this test was set to 22°–44°. The corresponding explosion distance can be obtained in accordance with the geometric relationship. The effects of the explosion-proof wall on shock wave diffraction under five proportional explosion distances and diffraction angles were calculated via numerical simulation. The specific layout is provided in Table 8.

**Table 8.** Relationship between diffraction angle and explosion distance.

| Test Group | Proportional Distance/m·g$^{-1/3}$ | Burst Distance/m | Diffraction Angle/° |
|---|---|---|---|
| 1-1 | 0.81 | 0.93 | 44 |
| 1-2 | 1.01 | 1.15 | 38 |
| 1-3 | 1.26 | 1.44 | 32 |
| 1-4 | 1.48 | 1.69 | 28 |
| 1-5 | 1.95 | 2.23 | 22 |

#### 4.2. Material Model

One of the simplest forms of an equation of state is that for an ideal polytropic gas which may be used in many applications involving the motion of gases. The state equation of air is approximated using the ideal gas state equation, and the specific expression is as follows:

$$P = (\gamma - 1)\frac{\rho}{\rho_0}E_0 \tag{4}$$

where $p$ is the air pressure, kPa; $\rho$ is the air density after compression or expansion, g/cm$^{-3}$; $\rho_0$ is the initial air density, g/cm$^{-3}$; $\gamma$ is the adiabatic index; and $E_0$ is the initial specific internal energy of air, kJ/m$^3$. The main parameters of air are shown in Table 9.

**Table 9.** Parameters for air.

| Parameter | Density/g·cm$^{-1/3}$ | Adiabatic Exponent | Specific Heat/J·(kg·K)$^{-1}$ | Specific Internal Energy/J·kg$^{-1}$ |
|---|---|---|---|---|
| Value | 0.001225 | 1.4 | 717.6 | $2.068 \times 10^5$ |

TNT was described using the JWL state equation, and its specific form is as follows:

$$P = C_1\left(1 - \frac{\omega}{r_1 v}\right)e^{-r_1 v} + C_2\left(1 - \frac{\omega}{r_2 v}\right)e^{-r_2 v} + \frac{\omega E_0}{v} \tag{5}$$

where $P$ is the pressure, kPa; $e$ is the internal energy of the explosive, kJ/m$^3$; $v$ is the relative volume of the explosive, cm$^3$/g; $E_0$ is the initial internal energy, kJ/m$^3$; and $C_1$, $r_1$, $C_2$, $r_2$, and $\omega$ are the five constants of the state equation. The main parameters of TNT are shown in Table 10.

**Table 10.** Parameters for TNT.

| Parameter | Density/g·cm$^{-1/3}$ | $C_1$/GPa | $C_2$/GPa | $r_1$ | $r_2$ | $\omega$ | VOD/m·s$^{-1}$ | $E_0$/J·m$^{-3}$ | $P_{CJ}$/GPa |
|---|---|---|---|---|---|---|---|---|---|
| Value | 1.63 | $3.73 \times 10^2$ | 3.74 | 4.15 | 0.9 | 0.35 | 6930 | $6 \times 10^9$ | 21 |

Sand was described using the compaction nonlinear state equation, the Drucker–Prager Strength Model, and the Tensile Stress Failure Model for Blast Resistance Simulation of Blast Walls. The material parameters are listed in Table 2. The specific form is as follows:

$$K(\rho) = \rho\frac{dP}{d\rho} \tag{6}$$

where $P$ is the current pressure, kPa; and $\rho$ is the density of the material under zero pressure, $g/cm^3$. The main parameters of sand are shown in Table 11 and the relationship between sand pressure change and density is shown in Figure 13.

**Table 11.** Parameters for sand.

| Parameter | Density/g·cm$^{-1/3}$ | Shear Modulus/MPa | Hydro Tensile Limit/KPa |
|---|---|---|---|
| Value | 2.028 | 76.9 | — |

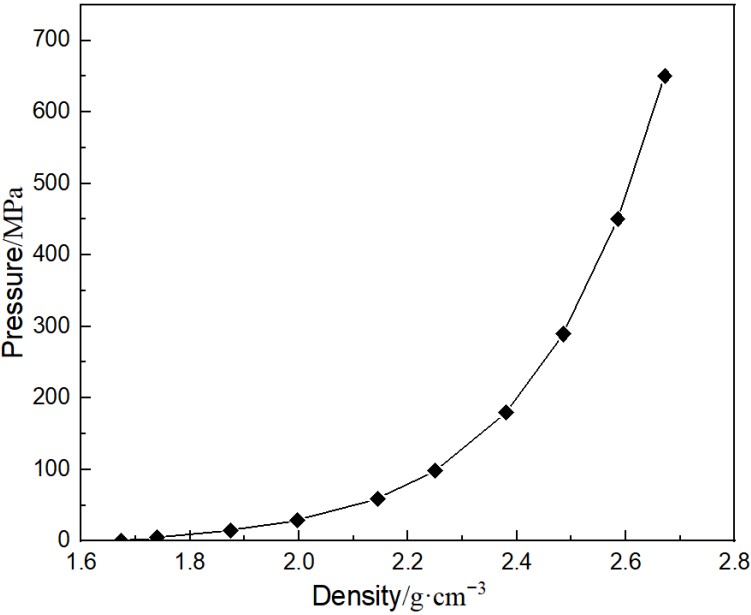

**Figure 13.** The relationship between sand density and pressure change.

*4.3. Model Size and Meshing*

In accordance with the design model size of the field experiment, the size of the air model was 22 m × 10 m × 5 m, the size of the sand model was 1 m × 5 m × 2.1 m, and the height of the medicine bag was 1.2 m above the ground. The TNT equivalent was consistent with the physical experiment. The air grid was 5 cm, and the explosion-proof wall grid was 10 cm. The 1/2 model was used for calculation because of the large model and its symmetry. The established model is shown in Figure 14.

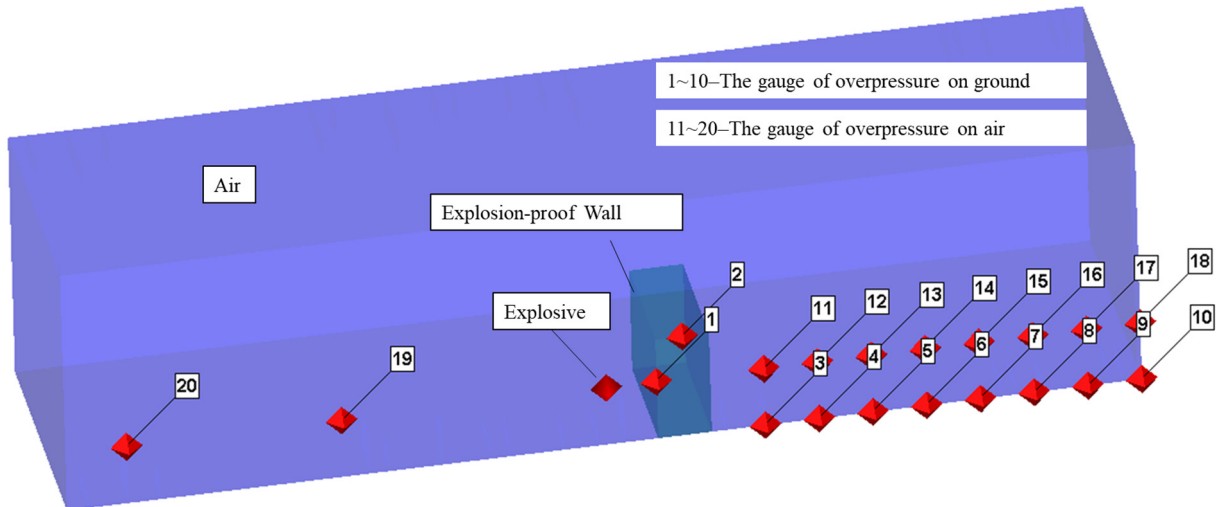

**Figure 14.** Establishment of the computational model.

To save calculation time and improve calculation accuracy, the unique remap technology of Autodyn software (2019R3).was used. This technology maps the calculation results obtained after a certain period in 1D and 2D onto another 3D model. This process not only considerably reduces calculation time, but also ensures the accuracy of the calculation results of the explosion problem. The specific process is illustrated in Figure 15. In the simulation process adopted in the current study, the remap mapping range was 90 cm, and the grid size was 0.1 cm.

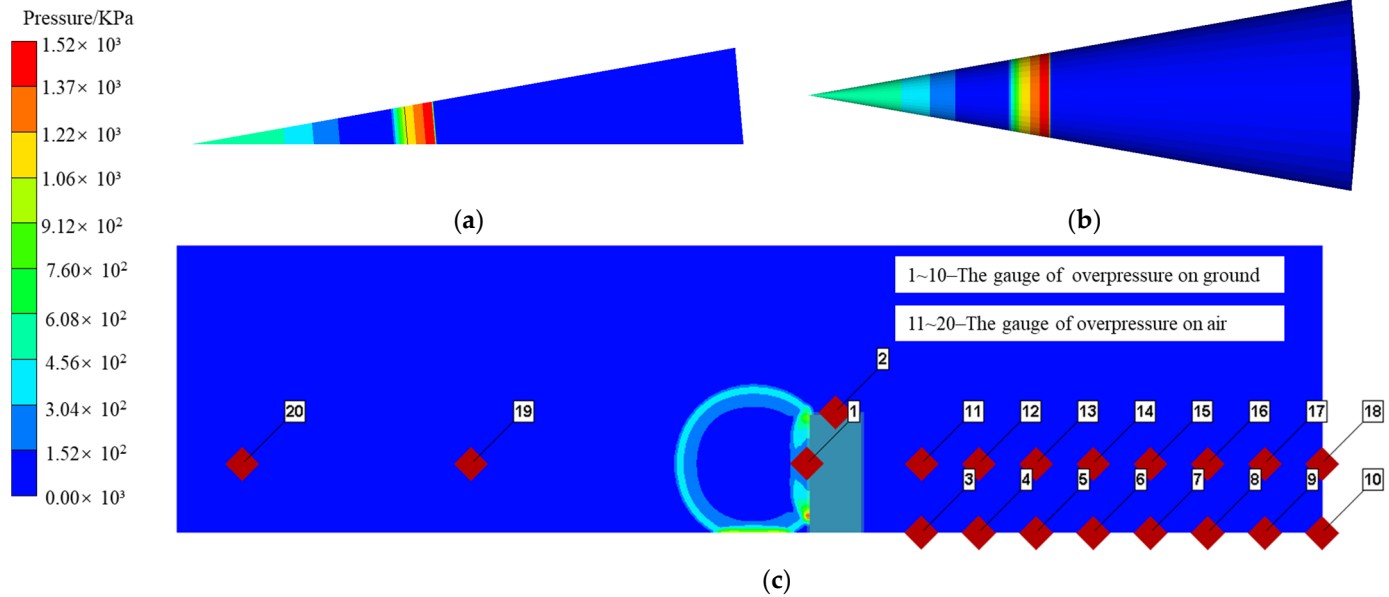

**Figure 15.** Remap mapping technology: (**a**) 1D explosion wave propagation; (**b**) 2D explosion wave propagation; (**c**) remap onto a 3D model.

### 4.4. Calculation Results and Analysis

Figure 16 shows the pressure cloud diagram of the shock wave diffracted jet flow field at different moments for five diffraction angles. At 0.5 ms, the two groups with a closer explosion distance had come into contact with the explosion-proof wall and formed a reflected wave. The explosion shock wave at 1–2 ms was reflected on the ground to form a Mach wave and advanced toward the explosion-proof wall. The reflection of the explosion-proof wall formed a downward and upward Mach wave. The downward Mach wave, the Mach wave formed by ground reflection, and the incident wave formed a superimposed wave at the wall foot, indicating that the wall foot was subjected to considerable shock wave pressure and continuous dynamic pressure. At 2.5–4.5 ms, the incident wave of the explosion shock wave was less than the reflected wave pressure formed on the wall. The incident wave continued to propagate forward along the top of the explosion-proof wall, and the sparse wave flowed to the high-pressure area, forming a cyclone on the top of the wall. The diffraction wave continued to propagate forward due to its interaction with the incident wave. At 12.5 ms, the diffracted wave was also reflected when it finally came into contact with the ground behind the wall and formed a Mach wave, which continued to propagate until it dissipated. The sequence of the main explosion shock wave on the front face of the explosion-proof wall is as follows: (1) incident wave of the explosion shock wave, (2) Mach wave reflected from the ground, and (3) superimposed wave of the Mach wave at the foot of the wall formed by the downward Mach wave reflected by the wall and the ground reflected wave. An inference can be made that if the explosion-proof wall is sufficiently far from the explosion source, then only the Mach wave reflected by the ground can reach the explosion-proof wall.

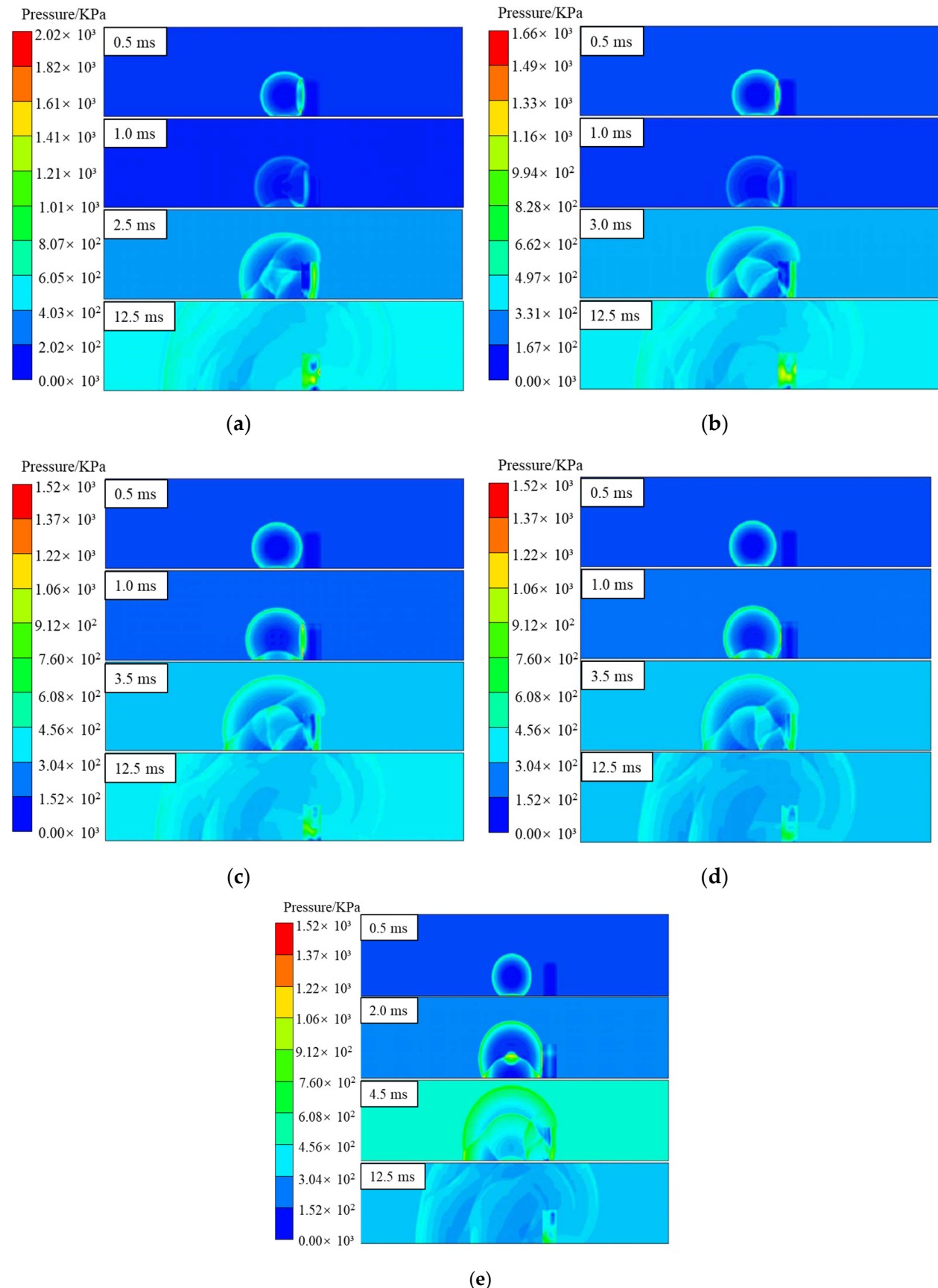

**Figure 16.** Pressure cloud of different diffraction angles: (**a**) 44°; (**b**) 38°; (**c**) 32°; (**d**) 28°; (**e**) 22°.

Figure 17 presents a comparison between the experimental and numerical simulation results when the diffraction angle was 32°. The simulation results are in good agreement with the experimental results, as shown in Figure 17. The experiment was performed on grassland and not on rigid ground, while the numerical calculation was conducted under ideal conditions without external interference and on rigid ground. An excessively large overpressure peak calculated during the simulation is normal. The pressure time history curve obtained by the experiment and the simulation indicated that the number and trend of each pressure peak were relatively consistent and could be used as the result of the trend analysis. The incident and reflected shock waves were superimposed significantly on the back side of the blast wall, and the peak pressure of the shock wave pressure behind the blast wall was tested at about 91% to 92% of the numerically calculated value, as shown in Figure 17.

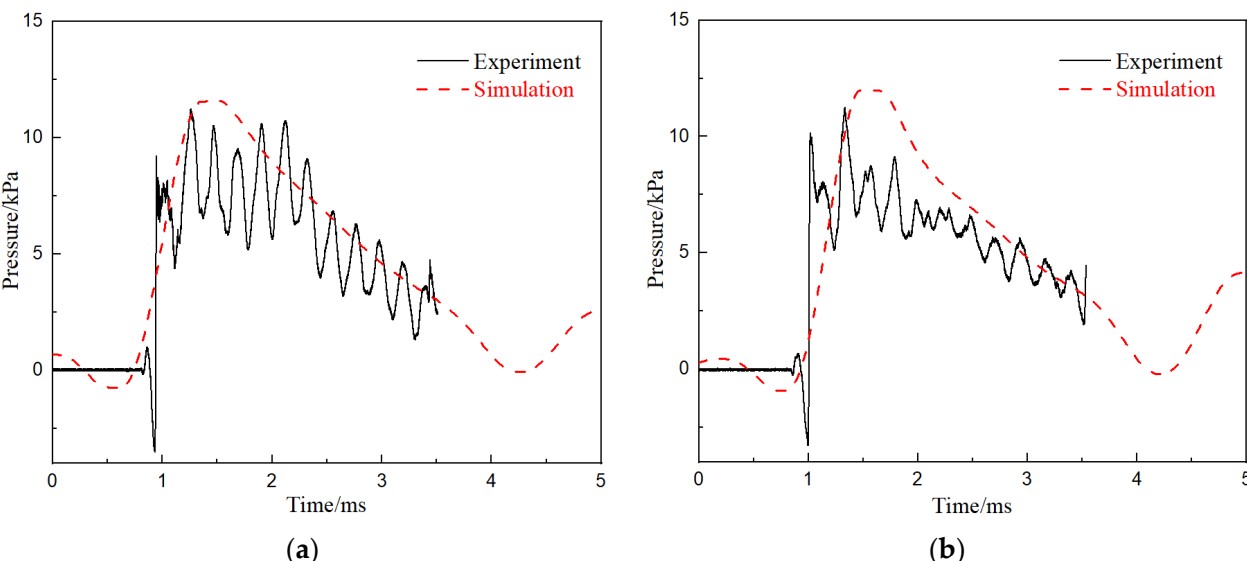

**Figure 17.** Comparison between the experiment and numerical simulation: (**a**) Point 4 and (**b**) Point 5.

When the explosive shock wave impacted the explosion-proof wall with different diffraction angles, the shock wave would generate varying pressures at different positions of the explosion-proof wall. The overpressure peak values calculated at different locations are provided in Table 12. A steel wire skeleton geotextile pocket was not added to the sand wall. Therefore, compactness was reduced and part of the pressure penetrated the sand wall, resulting in a small overpressure value at measuring point 1, but it minimally affected the study of the law of reflection overpressure at other measuring points behind the wall. The reduction of the shock wave by the explosion-proof wall is shown in Figure 18 and Table 13 (shock wave reduction $Q_1 = P_{19} - P_{13}$, shock wave reduction $Q_2 = P_{20} - P_{17}$, shock wave reduction rate $\eta_1 = (P_{19} - P_{13})/P_{19}$, shock wave reduction rate $\eta_2 = (P_{20} - P_{17})/P_{20}$, where the distances from $P_{19}$ and $P_{13}$ and from $P_{20}$ and $P_{17}$ to the explosion source were equal). This type of explosion-proof wall exhibits a good protection effect on the explosion shock wave. The reduction rate was more than 70% at the proportional distance position of the set measuring point.

**Table 12.** Peak pressure of test points in the numerical simulation.

| Measuring Points | Peak Pressure/kPa | | | | |
|---|---|---|---|---|---|
| | **44°** | **38°** | **32°** | **28°** | **22°** |
| 1 | 2540.04 | 1680.39 | 987.25 | 604.49 | 379.98 |
| 2 | 110.49 | 110.58 | 103.75 | 94.64 | 74.99 |
| 3 | 11.77 | 11.47 | 10.85 | 10.34 | 9.32 |
| 4 | 12.72 | 12.21 | 11.58 | 10.95 | 9.60 |
| 5 | 13.28 | 12.68 | 11.99 | 11.33 | 9.92 |
| 6 | 13.41 | 12.80 | 12.11 | 11.43 | 10.08 |
| 7 | 13.04 | 12.46 | 11.90 | 11.26 | 10.03 |
| 8 | 12.20 | 11.69 | 11.31 | 10.76 | 9.70 |
| 9 | 11.22 | 10.77 | 10.46 | 10.08 | 9.23 |
| 10 | 9.73 | 9.45 | 9.38 | 9.20 | 8.57 |
| 11 | 12.06 | 11.51 | 10.53 | 9.65 | 8.13 |
| 12 | 12.57 | 12.06 | 11.20 | 10.48 | 8.99 |
| 13 | 11.11 | 10.66 | 10.07 | 9.55 | 8.45 |
| 14 | 9.63 | 9.22 | 8.83 | 8.45 | 7.61 |
| 15 | 8.29 | 7.94 | 7.70 | 7.45 | 6.81 |
| 16 | 7.17 | 6.87 | 6.73 | 6.56 | 6.09 |
| 17 | 6.29 | 6.06 | 5.98 | 5.86 | 5.50 |
| 18 | 5.15 | 4.99 | 4.98 | 4.95 | 4.73 |
| 19 | 65.37 | 60.02 | 54.73 | 51.14 | 44.06 |
| 20 | 24.62 | 23.84 | 22.53 | 21.84 | 20.06 |

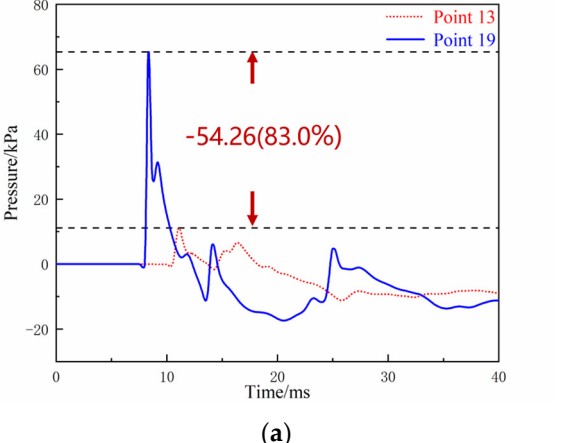
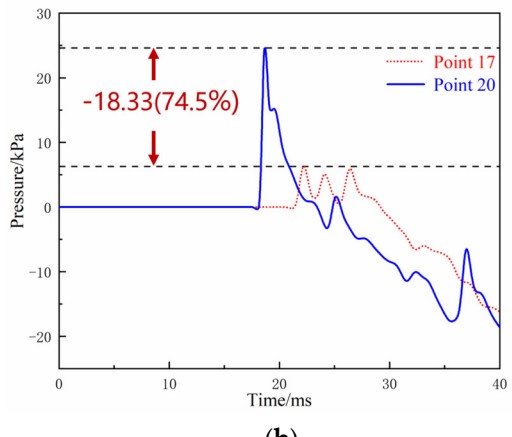

(**a**)        (**b**)

**Figure 18.** Shock wave attenuation of the explosion-proof wall with a diffraction angle of 44°: (**a**) shock wave attenuation 1; (**b**) shock wave attenuation 2.

**Table 13.** Shock wave attenuation.

| Diffraction Angle | $Q_1$/kPa | $\eta_1$/% | $Q_2$/kPa | $\eta_2$/% |
|---|---|---|---|---|
| 44° | 54.26 | 83.0 | 18.33 | 74.5 |
| 38° | 49.36 | 82.2 | 17.78 | 74.6 |
| 32° | 44.66 | 81.6 | 16.55 | 73.5 |
| 28° | 41.59 | 81.3 | 15.98 | 73.2 |
| 22° | 35.61 | 80.8 | 14.56 | 72.6 |

Figure 19 shows that shock wave pressure changed at different positions behind the wall at the five diffraction angles. The maximum overpressure of the ground measuring point behind the explosion-proof wall appeared at measuring point 6 (approximately two times the height of the explosion-proof wall), which initially increased and then decreased, with two times the height of the explosion-proof wall as the apex. It was similar to the linear decrease of the quadratic function. The maximum overpressure of the aerial measuring

point behind the explosion-proof wall appeared at measuring point 12 (approximately one times the height of the explosion-proof wall). After this distance, the quasi-linearity of the overpressure value continued to decrease. As the diffraction angle decreased, the pressure of the shock wave diffracted behind the wall also decreased. From Figure 19, a prediction can be made that as the measuring point moves farther away from the explosion-proof wall, the overpressure peaks will overlap regardless of whether the measuring point is in the air or on the ground. Therefore, shock wave pressure at different positions behind the wall was fitted.

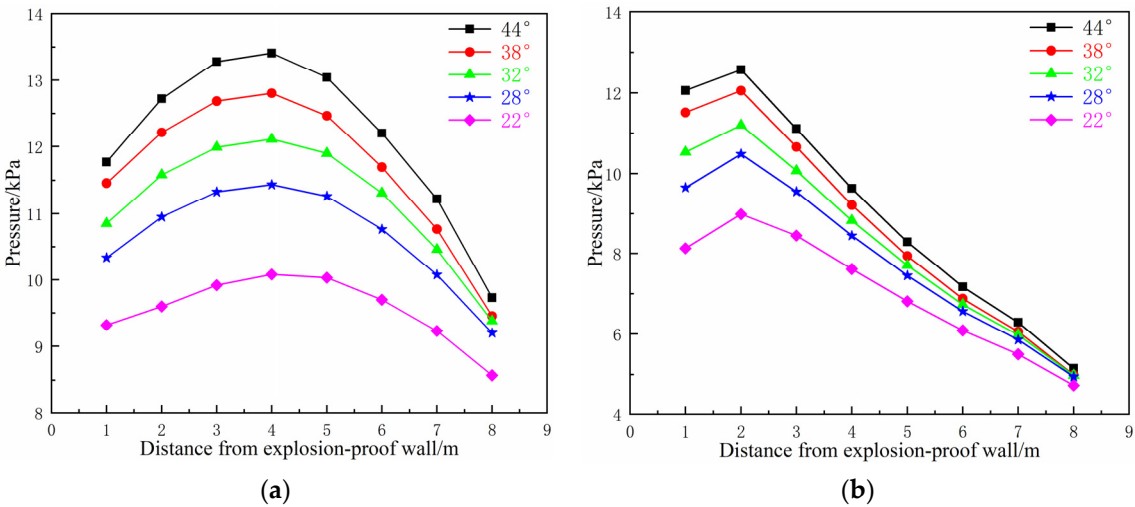

**Figure 19.** Shock wave pressure at different positions behind the wall: (**a**) ground behind the wall; (**b**) air behind the wall.

By fitting the shock wave pressure at different positions on the ground behind the explosion-proof wall via a quadratic polynomial as shown in Figure 20a, the prediction formula (7) of overpressure position formed by the explosion shock wave with different diffraction angles on the ground behind the explosion-proof wall was obtained. By plotting the obtained overpressure prediction formula as shown in Figure 20b, most curves will intersect at 8.29–9.40 m behind the wall (approximately 4–4.5 times the height of the explosion-proof wall). Regardless of how the diffraction angle of the shock wave and the explosion proportion distance change, the pressure here will be roughly the same. Thereafter, shock wave pressure decreased rapidly. It can be considered a relatively safe position, which we call a relatively safe area, and the place that continues to remain far from the explosion source can acquire higher security.

$$y_1 = ux^2 + wx + z \tag{7}$$

where $y_1$ is the peak value of overpressure at $x$ between the ground measurement point and the back surface of the wall under the current shock wave diffraction angle and proportional distance, kPa; $x$ is the distance from the measuring point to the back surface of the wall, m; and $u$, $w$, and $z$ are the constants related to the shock wave diffraction angle and the position of the measuring point, $-0.20387 \leq u \leq -0.09137$, $0.72935 \leq w \leq 1.53256$, and $8.60411 \leq z \leq 10.47339$.

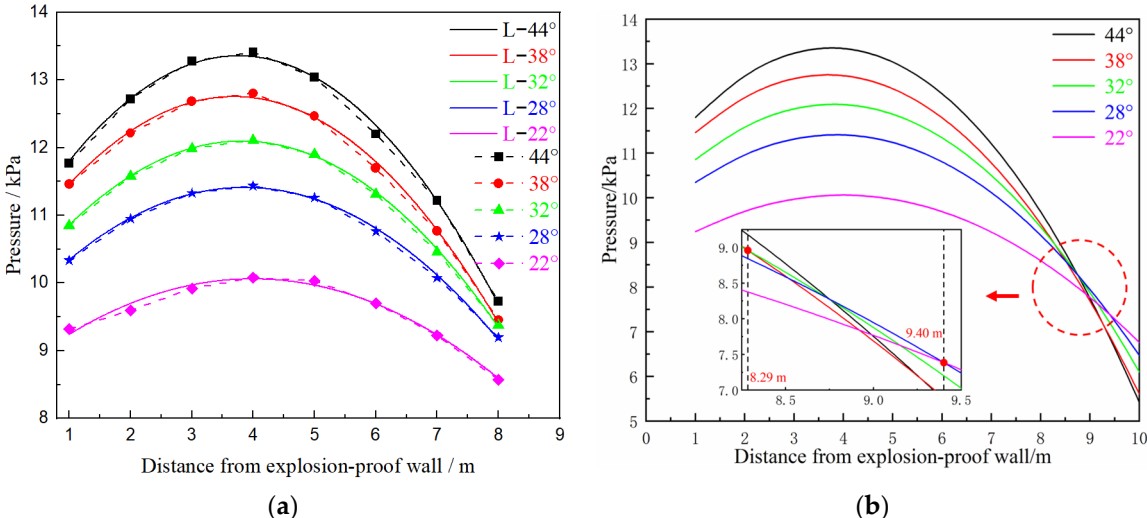

**Figure 20.** Safety zone prediction: (**a**) polynomial fitting; (**b**) safety point.

Linear fitting was performed for the points in the air of the explosion-proof wall after one times the height of the explosion-proof wall, as shown in Figure 21a. The prediction formula (8) of the overpressure position formed by the explosion shock wave with different diffraction angles on the ground behind the explosion-proof wall was obtained. By plotting the obtained overpressure prediction formula as shown in Figure 21b, most of the curves intersected at 7.42–8.60 m behind the wall (approximately 3.5–4 times the height of the explosion-proof wall). Similarly, it is called a relatively safe area.

$$y_2 = kx + h \tag{8}$$

where $y_2$ is the peak value of overpressure at $x$ between the air measurement point and the back surface of the wall under the current shock wave diffraction angle and proportional distance, kPa; $x$ is the distance from the measuring point to the back surface of the wall, m; and $k$ and $h$ are the constants related to the shock wave diffraction angle and the position of the measuring point, respectively, with $-1.22714 \leq k \leq -0.72143$ and $10.49000 \leq k \leq 14.73714$.

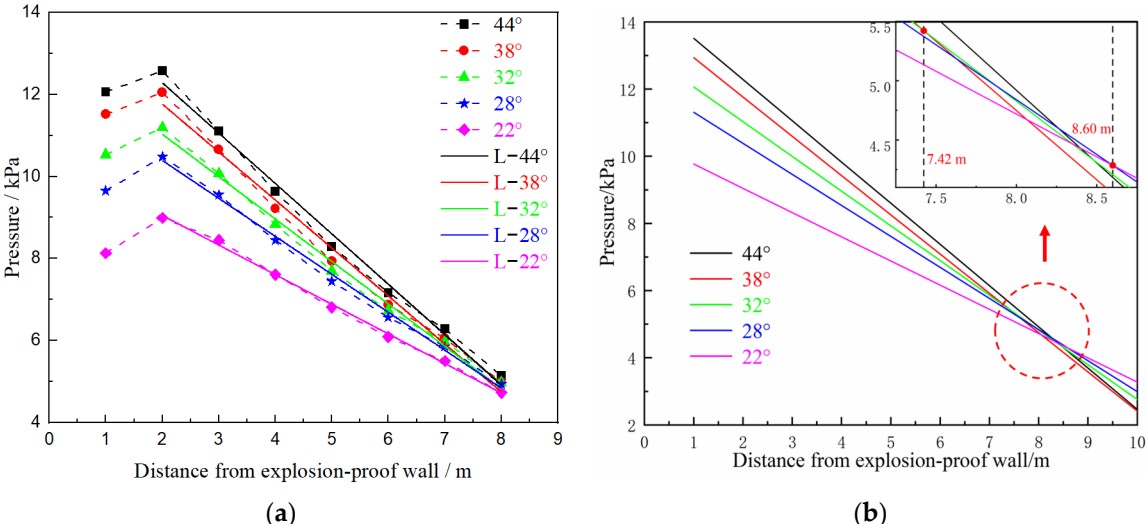

**Figure 21.** Safety zone prediction: (**a**) polynomial fitting; (**b**) safety point.

## 5. Conclusions

In the current study, an explosion-proof wall was set up to test a 1.5 kg TNT aerial explosion and perform a numerical simulation of the diffraction angle change. The conclusions drawn are as follows.

(1) When the diffraction angle was 32°, the maximum overpressure on the ground and in the air was approximately 2.4 times the height of the explosion-proof wall. When the diffraction angle was 28°, the maximum diffraction overpressure was approximately 2.4–2.6 times the height of the explosion-proof wall from the explosion-proof wall, and the maximum reflection overpressure was approximately 1.5 times the height of the explosion-proof wall from the explosion-proof wall. Overpressure behind the wall initially increased and then decreased. The greater the diffraction angle, the faster the attenuation speed of the diffraction overpressure in the air behind the wall increased with distance from the explosion-proof wall.

(2) Two evident overpressure peaks occurred in the pressure time history curve of the explosion shock wave. The time between the two pressure peaks $\Delta t$ is a key factor that affects the destructive effect of the explosion shock wave. If $\Delta t$ is less than the normal phase time $T^+$ in the pressure time history curve of the diffracted shock wave, then the damage capability of the shock wave will be significantly improved. The impulses of the two overpressure peaks rise rapidly, and the impulses are superimposed rapidly, exhibiting a strong destructive effect.

(3) By using the method for calculating the typical shock wave pressure in the ideal state, the peak value of dynamic pressure at three measuring points on the blast face was approximately two times that of overpressure. The air molecules behind the shock wave front possessed inertia, and dynamic pressure would have a longer positive duration than overpressure.

(4) By using Autodyn software to simulate different diffraction angles of the shock wave, the maximum overpressure of the ground measuring points behind the explosion-proof wall appeared at approximately two times the height of the explosion-proof wall, and the maximum overpressure of the air measuring points behind the explosion-proof wall appeared at approximately one times the height of the explosion-proof wall. The overpressure prediction formula of the air and ground behind the wall was obtained by fitting shock wave pressure at different positions behind the wall. The relative safety area of the ground behind the wall was approximately 4–4.5 times the height of the explosion-proof wall. The relative safety area of the air behind the wall was approximately 3.5–4 times the height of the explosion-proof wall.

(5) The relative safety area on the ground behind the wall is about 4 to 4.5 times the height of the blast wall, and the relative safety area in the air behind the wall is about 3.5 to 4 times the height of the explosion-proof wall.

**Author Contributions:** D.X. (Conceptualization, methodology, Writing—original draft, Writing—review & editing, Funding acquisition, Investigation, Supervision); W.Y. (Conceptualization, Formal analysis, methodology, Writing—original draft, Investigation); M.L. (methodology, software, Resources, Project administration); X.L. (Xiaoming Lü) (software, Project administration); K.L. (data curation, Visualization); J.Z. (software); X.L. (Xiaoshuang Li) (data curation); Y.L. (Visualization). All authors have read and agreed to the published version of the manuscript.

**Funding:** This research was funded by the Open Foundation of the Shaanxi Key Laboratory of Safety and Durability of Concrete Structures (No. SZ02201).

**Institutional Review Board Statement:** Not applicable.

**Informed Consent Statement:** Informed consent was obtained from all subjects involved in the study.

**Data Availability Statement:** Not applicable.

**Acknowledgments:** The authors would like to acknowledge the Shock and Vibration of Engineering Materials and Structures Key Laboratory of Sichuan Province and the Shaanxi Key Laboratory of Safety and Durability of Concrete Structures.

**Conflicts of Interest:** The manuscript has not been published before and is not being considered for publication elsewhere. We declare there is no conflict of interest.

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
