# Peer review of "Effects of an Explosion-Proof Wall on Shock Wave Parameters and Safe Area Prediction"

_sustainability, doi:10.3390/su151411164_

Round 1

Reviewer 1 Report

The manuscript provides a concise overview of the study conducted on the influences of an explosion-proof wall on shock wave parameters. The main findings are presented in a straightforward manner, describing the attenuation effect of the wall on the shock wave and the characteristics of shock wave impulse and dynamic pressure. However, there is room for improvement in terms of providing more specific details about the experimental setup, methodology, and results. The text could benefit from including information such as the dimensions and materials of the explosion-proof wall, the specific diffraction angles tested, and any specific measurement techniques employed to gather the data. Additionally, elaborating on the practical implications of the findings and their significance in terms of improving safety measures in explosive environments would enhance the overall impact of the text. Consider revising the text to provide a more comprehensive understanding of the experiment and its implications, ensuring that the information is clear, concise, and organized in a logical manner. Add clear figures for example Figure 2 is not clear with text, which is not readable. Add the theory of explosions, air pressure, and vibrations. Reduce the number of figures to less than 8. Add details of numerical simulation, and boundary conditions, and justify how it represents actual conditions. Add future work and recommendations of this research. 

Author Response

From Dingjun Xiao* 1,2, Wentao Yang 2,3 , Moujin Lin2, Xiaoming Lü4, Kaide Liu1, Jin Zhang2, Xiao shuang Li2 and Yu Long2

1   Shaanxi Key Laboratory of Safety and Durability of Concrete Structures;[email protected](dingjun xiao), [email protected](Kaide Liu)

2   School of Environment and Resource, Southwest University of Science and Technology, Mianyang 621010, China;[email protected](dingjun xiao), [email protected](Moujin Lin), [email protected](Jin Zhang), [email protected](Xiaoshuang Li), [email protected](Yu Long)

3   Chengdu Institute of Urban Safety and Emergency Management,Chengdu 610011, Sichuan, China, [email protected](Wentao Yang)

4   Ordnance Technology Research Institute, Ordnance Engineering College, Shijiazhuang 050003, China;[email protected](Xiaoming Lü)

To: Francoise Fu, Ph.D.

Editor-in-Chief

Sustainability

Re: sustainability-2464837: Effects of an Explosion-proof Wall on Shock Wave Parameters and Safe Area Prediction

Thank you very much for giving us this opportunity, and thank the reviewers for their correction and great suggestions which are all valuable and very helpful.

We would like to make a point-by-point response to the comments and suggestions in the following. We have fully addressed each concern and hope that this revised manuscript is now acceptable each concern is discussed in detail below.

Reviewer #1

Q1. The text could benefit from including information such as the dimensions and materials of the explosion-proof wall, the specific diffraction angles tested, and any specific measurement techniques employed to gather the data.

Reply: Thanks for the suggestion. We make the changes as follows:

The experimental explosion-proof unit consisted of five individual sand-filled blast walls with a single blast wall geometry of 1m x 1m x 2.1m.

Q2. Additionally, elaborating on the practical implications of the findings and their significance in terms of improving safety measures in explosive environments would enhance the overall impact of the text.

Reply: Thanks for the suggestion. In the conclusion section, we have added a fifth point as follows:

(5) The relative safety area on the ground behind the wall is about 4 to 4.5 times the height of the blast wall, and the relative safety area in the air behind the wall is about 3.5 to 4 times the height of explosion-proof wall.

Reviewer 2 Report

This manuscript present a study on the influences of an explosion-proof wall on shock wave. It is a well-writtem and can be accepted after mionor revision:

(1)  The parameters of material model is not given in the manuscript.

(2) Would you please give more illustartion on the design of explosion-proof wall.

The quality of English is OK.

Author Response

From Dingjun Xiao* 1,2, Wentao Yang 2,3 , Moujin Lin2, Xiaoming Lü4, Kaide Liu1, Jin Zhang2, Xiao shuang Li2 and Yu Long2

1   Shaanxi Key Laboratory of Safety and Durability of Concrete Structures;[email protected](dingjun xiao), [email protected](Kaide Liu)

2   School of Environment and Resource, Southwest University of Science and Technology, Mianyang 621010, China;[email protected](dingjun xiao), [email protected](Moujin Lin), [email protected](Jin Zhang), [email protected](Xiaoshuang Li), [email protected](Yu Long)

3   Chengdu Institute of Urban Safety and Emergency Management,Chengdu 610011, Sichuan, China, [email protected](Wentao Yang)

4   Ordnance Technology Research Institute, Ordnance Engineering College, Shijiazhuang 050003, China;[email protected](Xiaoming Lü)

To: Francoise Fu, Ph.D.

Editor-in-Chief

Sustainability

Re: sustainability-2464837: Effects of an Explosion-proof Wall on Shock Wave Parameters and Safe Area Prediction

Thank you very much for giving us this opportunity, and thank the reviewers for their correction and great suggestions which are all valuable and very helpful.

We would like to make a point-by-point response to the comments and suggestions in the following. We have fully addressed each concern and hope that this revised manuscript is now acceptable each concern is discussed in detail below.

Reviewer #2

Q1. The parameters of material model is not given in the manuscript

Reply: Thanks for the suggestion. We make the changes as follows:

One of the simplest forms of equation of state is that for an ideal polytropic gas which may be used in many applications involving the motion of gases.The state equation of air is approximated using the ideal gas state equation, and the specific expression is as follows:

                                  (4)

where p is the air pressure, kPa; ρ is the air density after compression or expansion, g/cm-3; ρ0 is the initial air density, g/cm-3; γ is the adiabatic index; and E0 is the initial specific internal energy of air, kJ/m3.

TNT was described using the JWL state equation, and its specific form is as follows:

                (5)

where P is the pressure, kPa; e is the internal energy of the explosive, kJ/m3; v is the relative volume of the explosive, cm3/g; E0 is the initial internal energy, kJ/m3; , , , , and are the five constants of the state equation.

Sand was described using the compaction nonlinear state equation. Drucker-Prager Strength Model and Tensile Stress Failure Model for Blast Resistance Simulation of explosion-proof wall.The material parameters are listed in Table 2. The specific form is as follows:

                                     (6)

where P is the current pressure, kPa; and ρ is the density of the material under zero pressure, g/cm3.

Q2. Would you please give more illustartion on the design of explosion-proof wall.

Reply: Thanks for the suggestion. We have added the the design of explosion-proof wall as follows:

The experimental explosion-proof unit consisted of five individual sand-filled blast walls with a single blast wall geometry of 1m x 1m x 2.1m.

Reviewer 3 Report

This work (2464837) may be of interests for readers of Sustainability. However, before the final decision being made, a major revision is required to provide a better and more rigorous work.

1.     The reason for the high impulse has not been analyzed, causing a lack of persuasiveness in subsequent research. Ground reflection should be considered and its impact on impulse should be discussed.

2.     In Fig 6 (a), y-axis and other graph ranges are not consistent, and the variation trend difference is not intuitive. The diagram should be improved or corrected consistently.

3.     In Fig 16, the comparison between numerical simulation results and experimental results is rarely discussed. It is recommended to conduct error analysis on the data to verify the reliability of numerical simulation.

4.     Many works cited by this article mainly introduced by the failure characteristics of Explosion-proof Wall. Moreover, theoretical models and methods should be introduced . A coupled thermal elastic plastic damage model for concrete subject to dynamic loading; A dynamic bounding surface plasticity damage model for rocks subjected to high strain rates and confinements; Implementation of Johnson-Holmquist-Beissel model in four-dimensional lattice spring model and its application in projectile penetration are recommended for reference.

Author Response

From Dingjun Xiao* 1,2, Wentao Yang 2,3 , Moujin Lin2, Xiaoming Lü4, Kaide Liu1, Jin Zhang2, Xiao shuang Li2 and Yu Long2

1   Shaanxi Key Laboratory of Safety and Durability of Concrete Structures;[email protected](dingjun xiao), [email protected](Kaide Liu)

2   School of Environment and Resource, Southwest University of Science and Technology, Mianyang 621010, China;[email protected](dingjun xiao), [email protected](Moujin Lin), [email protected](Jin Zhang), [email protected](Xiaoshuang Li), [email protected](Yu Long)

3   Chengdu Institute of Urban Safety and Emergency Management,Chengdu 610011, Sichuan, China, [email protected](Wentao Yang)

4   Ordnance Technology Research Institute, Ordnance Engineering College, Shijiazhuang 050003, China;[email protected](Xiaoming Lü)

To: Francoise Fu, Ph.D.

Editor-in-Chief

Sustainability

Re: sustainability-2464837: Effects of an Explosion-proof Wall on Shock Wave Parameters and Safe Area Prediction

Thank you very much for giving us this opportunity, and thank the reviewers for their correction and great suggestions which are all valuable and very helpful.

We would like to make a point-by-point response to the comments and suggestions in the following. We have fully addressed each concern and hope that this revised manuscript is now acceptable each concern is discussed in detail below.

Reviewer #3

Q1.The reason for the high impulse has not been analyzed, causing a lack of persuasiveness in subsequent research. Ground reflection should be considered and its impact on impulse should be discussed.

Reply: Thanks for the suggestion. We have added the analysis of the effect of ground reflection of shock waves on the impulse as follows:

Ground reflected waves on the impact of the explosion impulse can not be ignored, if located below the three-wave point traces, then the ground reflected waves will obviously affect the pressure peak, the explosion directly generated by the incident wave and the ground reflected waves will be superimposed and enhanced, if located above the three-wave point traces, the incident wave and the ground reflected waves will be significantly separated, weakening the pressure peak.

Q2. In Fig 6 (a), y-axis and other graph ranges are not consistent, and the variation trend difference is not intuitive. The diagram should be improved or corrected consistently.

Reply: Thanks for the suggestion. We make the changes as follows:

(a)                                   (b)

Fig. 6 Diffraction angle 28°: Overpressure distribution behind the wall: (a) Ground; (b) Air.

Q3. In Fig 16, the comparison between numerical simulation results and experimental results is rarely discussed. It is recommended to conduct error analysis on the data to verify the reliability of numerical simulation.

Reply: Thanks for the suggestion. We make the changes as follows:

Near the source of the experimental measurement points measured the peak pressure of the incident wave is about 64% of the peak pressure of the numerical calculation, the peak pressure of the reflected wave is about 67% of the peak pressure of the numerical calculation. Away from the source measurement point, the experimental measured peak pressure of the incident wave is about 76% of the peak pressure of the numerical calculation, the peak pressure of the reflected wave is about 92% of the peak pressure of the numerical calculation.

Q4. Many works cited by this article mainly introduced by the failure characteristics of Explosion-proof Wall. Moreover, theoretical models and methods should be introduced . A coupled thermal elastic plastic damage model for concrete subject to dynamic loading; A dynamic bounding surface plasticity damage model for rocks subjected to high strain rates and confinements; Implementation of Johnson-Holmquist-Beissel model in four-dimensional lattice spring model and its application in projectile penetration are recommended for reference.

Reply: Thanks for the suggestion. We make the changes as follows:

In terms of material type selection, porous brittle materials (e.g., porous rock or porous concrete) and foam materials have received attention in recent years in the field of protec-tion engineering, and studies have shown that in porous materials (porous rock [25], rigid polyurethane foam [26, 27], aluminum foam [28], etc.), the plastic compression process is the main feature of the plastic mechanical behavior of porous materials, and the pore col-lapse effect makes the attenuation of stress waves to be accelerated, thus providing a good seismic energy absorption effect.

[25]  Li Zhang, Li Chen, Qin Fang, et al. Mitigation of blast loadings on structures by an anti-blast plastic water wall[J]. Journal of Central South University, 2016,23(2):461-469.DOI:10.1007/s11771-016-3091-3.

[26] M. Barbier, D. Villamaina, E. Trizac. Microscopic origin of self-similarity in granular blast waves[J]. Physics of Fluids, 2016,28(8):83302.DOI:10.1063/1.4961047.

[27] Harold L. Brode. Numerical Solutions of Spherical Blast Waves[J]. Journal of Applied Physics, 2004,26(6):766-775.DOI:10.1063/1.1722085

[28] Aleem Ullah, Furqan Ahmad, Heung-Woon Jang, et al. Review of analytical and empirical estimations for incident blast pressure[J]. KSCE Journal of Civil Engineering, 2017,21(6):2211-2225.DOI:10.1007/s12205-016-1386-4.

Round 2

Reviewer 3 Report

This work (2464837) may be of interests for readers of Advances in Sustainability. However, before the final decision being made, a major revision is required to provide a better and more rigorous work.

1.     In 4.2 Material model, the material model parameters of the explosion-proof wall are relatively simple. Please use a table to list the specific parameters of the material model.

2.     In 4.4 Calculation results and analysis, the maximum error in the numerical simulation results of peak pressure is about 56%. If quantitative analysis of peak pressure is required in the article, the error between the simulation results and experimental results should be reduced.

3.     The suggested references have not been discussed, thus the discussion on theoretical work is not changed, which means that, without further revision this paper should be rejected next time.

Can be improved 

Author Response

Thank you very much for giving us this opportunity, and thank the reviewers for their correction and great suggestions which are all valuable and very helpful.

We would like to make a point-by-point response to the comments and suggestions in the following. We have fully addressed each concern and hope that this revised manuscript is now acceptable each concern is discussed in detail below.

Reviewer #3

Q1. In 4.2 Material model, the material model parameters of the explosion-proof wall are relatively simple. Please use a table to list the specific parameters of the material model.

Reply: Thanks for the suggestion. We have added the analysis of the Parameters for air,TNT and sand as follows:

Table 9 Parameters for air

Parameter

Density/g·cm−1/3

adiabatic exponent

specific Heat/J·(kgK)−1

specific internal energy/J·kg−1

Value

0.001225

1.4

717.6

2.068×105

Table 10 Parameters for TNT

Parameter

Density/g·cm−1/3

/GPa

/GPa

VOD/m·s−1

/J·m−3

/GPa

Value

1.63

3.73×102

3.74

4.15

0.9

0.35

6930

6×109

21

Table 11 Parameters for sand

Parameter

Density/g·cm−1/3

Shear modulus /MPa

Hydro Tensile Limit/KPa

Value

2.028

76.9

-1

Fig. 13 The relationship between sand density and pressure change

Q2.In 4.4 Calculation results and analysis, the maximum error in the numerical simulation results of peak pressure is about 56%. If quantitative analysis of peak pressure is required in the article, the error between the simulation results and experimental results should be reduced.

Reply: Thanks for the suggestion. We make the changes as follows:

The incident and reflected shock waves were superimposed significantly on the back side of the blast wall, and the peak pressure of the shock wave pressure behind the blast wall was tested at about 91% to 92% of the numerically calculated value, as shown in Fig.17.

(a)                              (b)

Fig. 17 Comparison between the experiment and numerical simulation: (a) Point 4 (b) Point 5

Q3. The suggested references have not been discussed, thus the discussion on theoretical work is not changed, which means that, without further revision this paper should be rejected next time.

Reply: Thanks for the suggestion. We make the changes as follows:

The choice of an appropriate material dynamic model has a significant impact on the results of numerical calculations. Ma[25] proposed an improved Johnson-Holmquist-Beissel model that can well predict the projectile penetration behavior of brittle materials. A new modeling strategy with inclusions and FDEM method is also proposed to study the dynamic response and fracture behavior of geomaterials[26]. In this paper, the compaction nonlinear state equation is used to describe the effect of blast walls in blast loading.

  1. Jianjun Ma, Junjie Chen, Junwei Guan, et al. Implementation of Johnson-Holmquist-Beissel model in four-dimensional lattice spring model and its application in projectile penetration[J]. International Journal of Impact Engineering, 2022:104340.DOI:https://doi.org/10.1016/j.ijimpeng.2022.104340.
  2. 26. Yuexiang Lin, Jianjun Ma, Zhengshou Lai, et al. A FDEM approach to study mechanical and fracturing responses of geo-materials with high inclusion contents using a novel reconstruction strategy[J]. Engineering Fracture Mechanics, 2023,282:109171.DOI:https://doi.org/10.1016/j.engfracmech.2023.109171.

Round 3

Reviewer 3 Report

This paper can be accepted.